# MEDFRAMEQA: A MULTI-IMAGE MEDICAL VQA BENCHMARK FOR CLINICAL REASONING

## ABSTRACT

Medical education videos capture the systematic, multi-image diagnostic reasoning that clinicians employ in practice—examining series of related scans, comparing views, and synthesizing findings across modalities. To evaluate whether MLLMs can perform this fundamental aspect of clinical reasoning, we introduce MEDFRAMEQA —the first benchmark explicitly designed to test multi-image medical VQA through educationally-validated diagnostic sequences. To build MEDFRAMEQA with high-scalability and high-quality, we develop 1) an automated pipeline that extracts temporally coherent frames from medical videos and constructs VQA items whose content evolves logically across images, and 2) a multiple-stage filtering strategy, including model-based and manual review, to preserve data clarity, difficulty, and medical relevance. The resulting dataset comprises 2,851 VQA pairs (gathered from 9,237 high-quality frames in 3,420 videos), covering nine human body systems and 43 organs; every question is accompanied by two to five images. We comprehensively benchmark 11 advanced Multimodal LLMs—both proprietary and open source, with and without explicit reasoning modules—on MEDFRAMEQA. The evaluation challengingly reveals that all models perform poorly, with most accuracies below 50%, and accuracy fluctuates as the number of images per question increases. Error analysis further shows that models frequently ignore salient findings, mis-aggregate evidence across images, and propagate early mistakes through their reasoning chains; results also vary substantially across body systems, organs, and modalities. These findings highlight a critical gap: while MLLMs may handle single-image medical tasks, they fail at the multi-image comparative reasoning that defines real clinical practice. We hope this work can catalyze research on clinically grounded, multi-image reasoning and accelerate progress toward more capable diagnostic AI systems.

## 1 INTRODUCTION

Multimodal Large Language Models (MLLMs) have quickly emerged as a powerful paradigm for enabling advanced AI systems in clinical and medical domains (Xie et al., 2025; OpenAI, 2023a; Li et al., 2023; Tu et al., 2023; Saab et al., 2024; Huang et al., 2025; Wu et al., 2025). In practice, clinicians frequently employ multi-image diagnostic workflows, comparing related scans and synthesizing findings across different views and time points. Current evaluation benchmarks, however, focus predominantly on isolated, single-image analysis, *e.g.*, (Lau et al., 2018; Ben Abacha et al., 2019; 2021; He et al., 2020; Liu et al., 2021; Zhang et al., 2023; Hu et al., 2024; Chen et al., 2024). The left panel of Figure 1 shows a typical SLAKE (Liu et al., 2021) example whose answer requires nothing more than basic object recognition in one frame. In everyday care, however, clinicians rarely rely on a lone snapshot; they routinely compare multiple images taken from different views, modalities, or time points before making a diagnosis.

Only recently has the vision community begun to tackle multi-image VQA. A handful of new benchmarks—such as Yue et al. (2024a;b); Zuo et al. (2025)—include questions that reference more than one picture. Yet their tasks still fall short of the integrative reasoning medicine demands, as the images from these benchmarks are typically treated as separate clues rather than as innately complementary pieces of a single, coherent scenario. The MedXpertQA example in the middle panel of Figure 1 illustrates this gap: the two images share no obvious physiological connection or causal chain, so it is possible for a model to still answer correctly without genuinely synthesizing

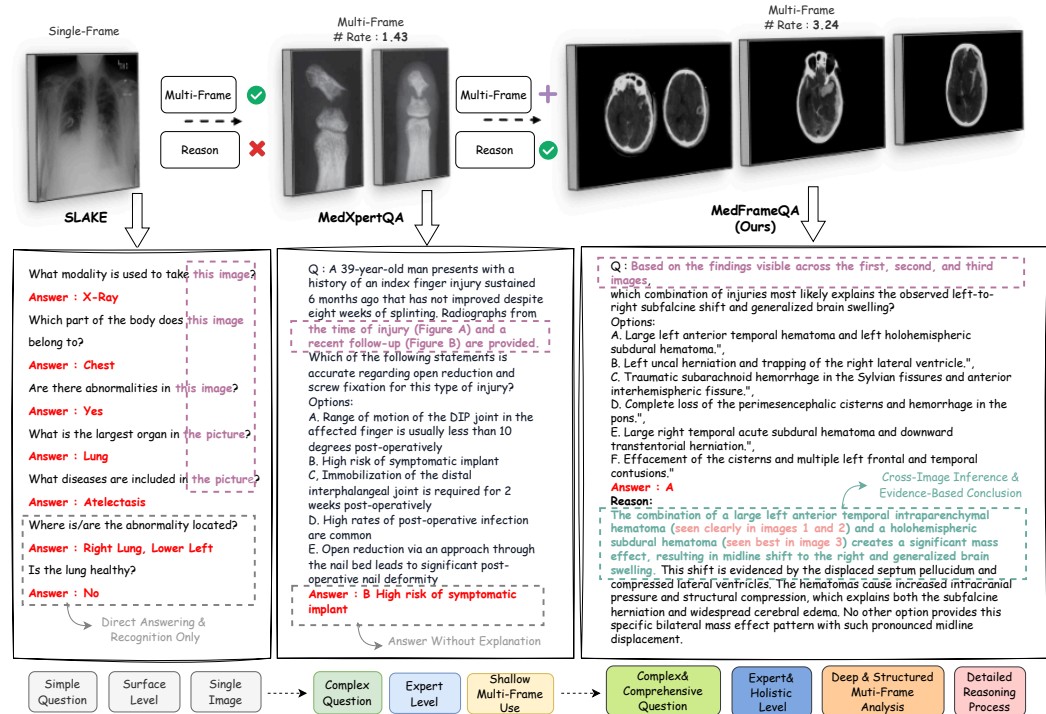

Figure 1: **Comparison of medical VQA benchmarks.** MEDFRAMEQA introduces multi-image, clinically grounded questions that require comprehensive reasoning across all images. Unlike prior benchmarks such as SLAKE (Liu et al., 2021) and MedXpertQA (Zuo et al., 2025), it emphasizes diagnostic complexity, expert-level knowledge, and explicit reasoning chains.

information from both. Consequently, success on such datasets therefore says little about a system's ability to perform the integrative, cross-image reasoning required for real diagnostic practice.

To bridge this gap, we introduce MEDFRAMEQA, the first benchmark explicitly designed to test multi-image reasoning in medical VQA by leveraging YouTube's rich repository of medical education videos (Osman et al., 2022; Akakpo and Akakpo, 2024). Our approach focuses on educational video sequences with temporally and semantically connected visual content that demonstrate diagnostic reasoning within coherent clinical presentations. Building on this insight and drawing inspiration from the prior work (Ikezogwo et al., 2023), we propose a VQA generation pipeline that automatically constructs multi-image VQA questions from keyframes extracted from 3,420 medical videos, spanning 9 human body systems and 43 organs across diverse anatomical regions. We first curated videos ranging from 5 minutes to 2 hours using 114 combinatorial search queries across imaging modalities and clinical findings. Keyframes were then extracted and rigorously filtered by GPT-4o for image quality, medical relevance, informativeness, and privacy. Audio narrations were transcribed, temporally aligned to frames within a defined margin, and refined by GPT-4o for clinical accuracy. Consecutive frame-caption pairs with a shared clinical focus were merged into multi-frame clips to preserve narrative continuity. GPT-4o then generated multiple-choice VQA items for each clip, requiring advanced clinical reasoning and multi-image analysis. A final two-stage filtering process—automated difficulty filtering via strong MLLMs and manual quality control—ensured a challenging, high-quality VQA benchmark tailored for medical imaging content.

This data curation pipeline yields MEDFRAMEQA, which consists of 2,851 challenging multi-image VQA questions requiring reasoning across temporally coherent sequences (2-5 frames each). These sequences include multi-view images of the same anatomy, progressive disease stages within educational narratives, and cross-modal comparisons—all derived from continuous educational video content rather than arbitrary image collections. As illustrated in the right panel of Figure 1, each item bundles a natural-language query with multiple frames, reducing reliance on single-image analysis. Moreover, we provide gold-standard rationales derived from source video transcripts, explicitly linking each image to the answer. We benchmark 11 state-of-the-art MLLMs on MEDFRAMEQA

Table 1: **Comparison of MEDFRAMEQA with Existing Benchmarks.** MEDFRAMEQA supports multi-image reasoning within real-world clinical video scenarios and paired reasoning across frames. The paired reasoning in MEDFRAMEQA is derived from the transcripts from original video clips.

| Benchmark | # Images | # Questions | # Rate | Multi-Image | Real World Scenarios | Paired Reasoning Across Multi Images |
|---|---|---|---|---|---|---|
| VQA-RAD (Lau et al., 2018) | 315 | 3515 | 0.09 | ✗ | ✗ | ✗ |
| VQA-Med-2019 (Ben Abacha et al., 2019) | 500 | 500 | 1.00 | ✗ | ✗ | ✗ |
| VQA-Med-2021 (Ben Abacha et al., 2021) | 500 | 500 | 1.00 | ✗ | ✗ | ✗ |
| PathVQA (He et al., 2020) | 858 | 6,719 | 0.13 | ✗ | ✗ | ✗ |
| SLAKE-En (Liu et al., 2021) | 96 | 1,061 | 0.09 | ✗ | ✗ | ✗ |
| PMC-VQA (Zhang et al., 2023) | 29,021 | 33,430 | 0.87 | ✗ | ✗ | ✗ |
| OmniMedVQA (Hu et al., 2024) | 118,010 | 127,995 | 0.92 | ✗ | ✗ | ✗ |
| GMAI-MMBench (Chen et al., 2024) | 21,180 | 21,281 | 1.00 | ✗ | ✗ | ✗ |
| MMMU (H&M) (Yue et al., 2024a) | 1,994 | 1,752 | 1.14 | ✓ | ✗ | ✓ |
| MMMU-Pro (H&M) (Yue et al., 2024b) | 431 | 346 | 1.25 | ✓ | ✗ | ✓ |
| MedXpertQA MM (Zuo et al., 2025) | 2852 | 2000 | 1.43 | ✓ | ✓ | ✗ |
| MEDFRAMEQA | **9237** | **2851** | **3.24** | ✓ | ✓ | ✓ |

and find that their accuracies mostly fall below 50% with substantial performance across different body systems, organs, and modalities, revealing critical gaps between current model capabilities and clinical diagnostic requirements, particularly in video-derived multi-image reasoning scenarios.

## 2 RELATED WORK

**Reasoning Multimodal Large Language Models** With advances in models and benchmarks, interest in the reasoning capabilities of MLLMs has grown (Wang et al., 2024; Xie et al., 2024; Chen et al., 2025; Deng et al., 2025). Recent MLLMs now support medical reasoning tasks like clinical decision-making, chain-of-thought generation, and diagnostic inference (AlSaad et al., 2024). Llava-Med (Li et al., 2023) and GPT-4V (OpenAI, 2023b) show generalist abilities in radiology and biomedical VQA but often lack interpretable reasoning. MedCoT (Wang et al., 2025) addresses this with a multi-expert prompting framework that improves rationale quality and accuracy. MedVLM-R1 (Pan et al., 2025) applies reinforcement learning to encourage plausible rationales without ground truth, improving radiology QA. Med-Gemini (Saab et al., 2024) combines domain-adapted prompting with long-context modeling for complex cross-modal inference. These advancements in applying MLLMs to medical reasoning tasks underscore the critical need for rigorous benchmarks that effectively evaluate their reasoning capabilities.

**Multimodal Medical Benchmarks** Existing benchmarks for evaluating MLLMs in the medical domain remain limited in scope. Most notably, the majority focus on single-image question answering tasks. For example, VQA-RAD (Lau et al., 2018), VQA-Med-2019 (Ben Abacha et al., 2019), VQA-Med-2021 (Ben Abacha et al., 2021), and SLAKE (Liu et al., 2021) primarily target single-question VQA tasks within the radiology domain, while Path-VQA (He et al., 2020) is dedicated exclusively to pathology. With the rapid advancement of MLLMs, more generalized benchmarks such as PMC-VQA (Zhang et al., 2023), OmniMedVQA (Hu et al., 2024), and GMAI-MMBench (Chen et al., 2024) have been introduced to assess broader model capabilities across diverse medical fields. However, these benchmarks remain limited, as they primarily focus on single-image VQA tasks—falling short of reflecting the demands of real-world medical applications. Recent efforts such as MMMU (H&M) (Yue et al., 2024a), MMMU-Pro (H&M) (Yue et al., 2024b), and MedXpertQA MM (Zuo et al., 2025) have incorporated multi-image VQA tasks. Nonetheless, their construction overlooks the critical need for clinical reasoning across multiple images—a core requirement in real-world diagnostic settings. Moreover, these VQA benchmarks lacks of ground-truth reasoning chains, making it difficult to determine whether the models are genuinely performing multi-image reasoning. We provide a comprehensive comparison of MEDFRAMEQA with existing benchmarks in Table 1.

**Video Data For Medical Benchmarking** Recent studies have advanced the use of video data for medical dataset construction. Speech recognition models like Whisper (Radford et al., 2023) have made it easier to extract data from videos (Zellers et al., 2021; Zhang et al., 2025). Quilt-1M (Ikezogwo et al., 2023) collected one million paired image-text samples from histopathology YouTube videos. MedVidQA (Gupta et al., 2023) and NurViD (Hu et al., 2023) target instructional and nursing procedures. Cotaract-1K (Ghamsarian et al., 2024) consists of 1,000 videos of cataract surgeries conducted in the eye clinic from 2021 to 2023. Despite advancements in video dataset construction, limited attention has been paid to leveraging video data for benchmarking MLLMs in the medical

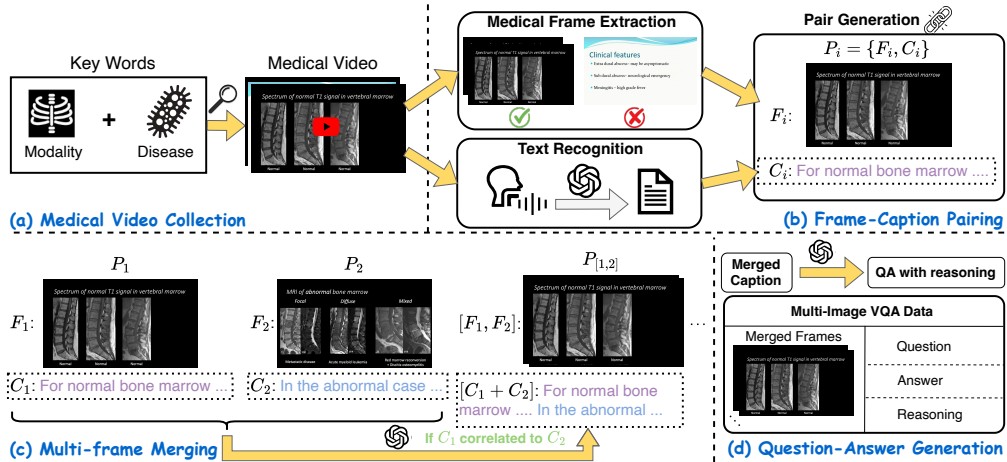

Figure 2: **Our data generation pipeline.** (a) Medical Video Collection: Collecting 3,420 medical videos via clinical search queries (Section 3.1). (b) Frame-Caption Pairing: Extracting keyframes and aligning with transcribed captions. (Section 3.2) (c) Multi-Frame Merging: Merging clinically related frame-caption pairs into multi-frame clips. (Section 3.3)(d) Question-Answer Generation: Generating multi-image VQA from the multi-frame clips. (Section 3.4)

domain. YouTube's rich medical content (Osman et al., 2022; Derakhshan et al., 2019) offers natural reasoning chains for multi-frame VQA evaluation. To this end, we utilize YouTube videos and design a VQA generation pipeline that automatically constructs multi-image VQA questions, aiming to assess the reasoning capabilities of MLLMs across complex multi-image scenarios.

# 3 MEDFRAMEQA BENCHMARK

## 3.1 MEDICAL VIDEO COLLECTION

As the first step in building MEDFRAMEQA, we assemble a large pool of clinically relevant videos from YouTube (illustrated in Figure 2(a)). Specifically, we curate 114 carefully designed search queries, each formed by pairing a common imaging modality (*e.g.* MRI, X-Ray, CT, and radiograph) with a frequently encountered disease or finding (*e.g.* brain tumor, pneumonia, chest, and bone fracture). This combinatorial list gives broad coverage of routine diagnostic scenarios; the full set of keywords is provided in Section D. Then, for every query, we retrieve the top results and discard clips shorter than 5 minutes or longer than 2 hours. The remaining corpus comprises 1,971 high-resolution, narration-rich medical videos that serve as the raw material for MEDFRAMEQA.

## 3.2 FRAME-CAPTION PAIRING

**Medical Frame Extraction.** To process the raw video collected, the first task is to identify the corresponding medical frames. Following Ikezogwo et al. (2023), we run FFmpeg (https://ffmpeg.org/) to extract key-frames—those delineating the scene boundaries and often indicating significant visual transitions—and record the corresponding temporal span of each segment ($f_{start}$, $f_{end}$). Each candidate frame is then evaluated by GPT-4o (Hurst et al., 2024) under four criteria: (1) *image quality*, evaluating the clarity and medical relevance of the frame; (2) *prominence of medical content*, determining if the frame predominantly consists of medical imagery; (3) *informative content*, checking if the frame is understandable and holds significant information; and (4) *privacy*, ensuring the frame excludes unrelated human faces, such as those of presenters in video conferences. Note that only frames satisfying all four requirements are retained. More details about the frame filtering criteria can be found in Section F.1.

This filtering step leaves us with a sequence of qualified key-frames and their temporal spans:

$$S_F = [F_1, \cdots F_m], \quad D_F = [(f_{start}^1, f_{end}^1), \cdots (f_{start}^m, f_{end}^m)], \tag{1}$$

where $m$ is the number of extracted medical frames. $S_F$ and $D_F$ are the sequence of frames and times, respectively.

**Text Recognition.** We next transcribe the audio track with Whisper (Radford et al., 2023). The model returns a sequence of $n$ text snippets and their time stamps:

$$S_T = [T_1, \cdots T_n], \quad D_T = [(t_{start}^1, t_{end}^1), \cdots (t_{start}^n, t_{end}^n)], \tag{2}$$

**Pair Generation.** Our third task now is to pair the medical frame with the corresponding caption. Intuitively, each frame can be simply paired with the text snippets that emerge concurrently with it during the same time interval. However, narration in medical videos can lag behind or precede the exact moment a frame is shown. To associate each frame ($F_i$) with all relevant speech, we define a symmetric margin ($\Delta$) seconds around the frame's interval and gather every transcript whose span intersects that window $\left[f_{start}^i - \Delta, f_{end}^i + \Delta\right]$. Then all snippets within this window range will be concatenated to form a coarse caption $\tilde{C}_i = \left[T_j, T_{j+1}, \ldots, T_k\right]$.

Then we leverage `GPT-4o` to enhance the quality of $\tilde{C}_i$. Specifically, `GPT-4o` is instructed to (i) remove statements unrelated to the displayed frame and (ii) refine the description to ensure the correct usage of clinical terminology. Formally,

$$C_i = \texttt{GPT-4o}\left(\tilde{C}_i, F_i \mid I_{rephrase}\right), \tag{3}$$

where $C_i$ denotes the refined caption, and $I_{rephrase}$ is the prompt (see Section F.1 for more details). The final frame–caption pair is $P_i = \{F_i, C_i\}$, and the sequence of frame-caption pairs of the entire video is $S_P = [P_1, \cdots, P_n]$.

### 3.3 MULTI-FRAME MERGING

The paired frames described above usually belong to longer narrative units within educational presentations—for example, a radiologist may spend several consecutive slides discussing the same lesion during a structured teaching session. To capture such continuity, we merge adjacent frame-caption pairs into multi-frame "clips" whenever their captions describe the same clinical concept within the educational context. The paired caption of each frame already provides a description of its visual content; hence, we rely entirely on the textual correlation between the captions to determine if there is a connection between two frames. Specifically, as illustrated in Figure 2(c), for every consecutive pair $P_i = \{F_i, , C_i\}$ and $P_{i+1} = \{F_{i+1}, , C_{i+1}\}$, we ask `GPT-4o` (prompt in Section F.2) whether these two captions are correlated. If yes, we then combine these two pairs: $P_{[i,i+1]} = \left\{[F_i, F_i + 1], [C_i \oplus C_{i+1}]\right\}$, where $\oplus$ represents the text concatenation. We then compare the merged caption $[C_i \oplus C_{i+1}]$ with the next caption $C_{i+2}$; if the relation persists, we append $P_{i+2}$ to the group. This sliding process continues until (i) the next caption is judged unrelated or (ii) the group reaches a maximum of five frames, the limit we adopt in this work.

Applying the above procedure to all videos yields 7,998 multi-frame clips, each containing 2–5 medically coherent frame-caption pairs. These clips constitute the basic building blocks for the subsequent VQA-item generation stage.

### 3.4 QUESTION ANSWERING GENERATION

As shown in Figure 2(d), for each merged group $P_{[i,i+1\cdots]} = \{[F_i, F_{i+1}, \cdots], [C_i \oplus C_{i+1}, \cdots]\}$, we instruct `GPT-4o` to generate challenging multiple-choice questions. Formally,

$$Q, A, R = \texttt{GPT-4o}\left([C_i \oplus C_{i+1} \cdots] \mid I_{gen}\right), \tag{4}$$

where $Q, A, R$ are the generated question, the correct answer, and the reasoning, respectively. $I_{gen}$ is the generation prompt, enforcing four requirements: (1) *Information Grounding*: all questions must rely solely on visual evidence explicitly described in the educational video captions; (2) *Educational Clinical Reasoning*: each question should probe skills demonstrated in medical education contexts such as anatomical localization and differential diagnosis within structured presentations; (3) *Contextual Interaction*: the wording must reference the images in order (*e.g.*, "in the first image ..., whereas in the third image ...") and require synthesizing information across the educational sequence; (4) *Distraction Options*: every item includes plausible but incorrect answer choices that differ from the ground truth in clinical details within the educational context. The complete $I_{gen}$ is provided in Section F.3. Lastly, each clip is packaged as $\{Q, A, R, [F_i, F_{i+1} \cdots]\}$, forming a single entry.

Table 2: **Accuracy of Models on MEDFRAMEQA.** We report the system-wise accuracy of models on MEDFRAMEQA. The results are averaged over all the tasks in MEDFRAMEQA. The best results on each system and average accuracy are highlighted in bold. In general, all assessed models demonstrate persistently low accuracy, with system-wise performance of substantial variability in task difficulty.

| Model | Accuracy per System | | | | | | | | | Avg |
|---|---|---|---|---|---|---|---|---|---|---|
| | CNS | RES | CIR | DIG | URI | REP | END | MSK | AUX | |
| *Proprietary Reasoning Models* | | | | | | | | | | |
| o1 | 46.91 | 48.88 | 49.49 | 47.45 | 49.03 | 42.26 | 47.68 | 51.59 | 48.75 | 47.91 |
| o3 | 47.81 | 52.00 | 50.00 | 48.48 | 50.71 | 45.02 | 51.84 | 54.90 | 50.41 | 50.18 |
| o4-mini | 46.03 | 49.78 | 48.74 | 48.63 | **51.85** | 43.62 | 52.44 | 53.38 | 50.82 | 49.40 |
| Gemini-2.5-Flash | 48.82 | **58.26** | **57.21** | **50.25** | 48.61 | **55.81** | **55.38** | **60.21** | **52.85** | **54.75** |
| Claude-3.7-Sonnet | 49.21 | 46.09 | 53.23 | **50.25** | 49.07 | 47.57 | 47.81 | 52.42 | 49.59 | 49.67 |
| *Open-Source Reasoning Models* | | | | | | | | | | |
| QvQ-72B-Preview | 44.88 | 46.67 | 47.43 | 41.13 | 45.68 | 47.00 | 47.68 | 49.37 | 47.15 | 46.44 |
| *Proprietary Non-Reasoning Models* | | | | | | | | | | |
| GPT-4o | 48.82 | 49.13 | 37.31 | 50.00 | 43.98 | 45.88 | 46.22 | 43.60 | 44.31 | 45.67 |
| GPT-4o-mini | 41.73 | 36.52 | 39.30 | 28.36 | 35.65 | 33.83 | 30.68 | 34.95 | 34.96 | 34.55 |
| GPT-4-Turbo-V | 45.28 | 46.09 | 42.79 | 49.75 | 43.06 | 48.63 | 49.80 | 45.16 | 46.75 | 46.69 |
| *Open-Source Non-Reasoning Models* | | | | | | | | | | |
| Qwen2.5-VL-72B-Instruct | 43.18 | 47.39 | 42.29 | 39.80 | 39.81 | 43.41 | 43.03 | 44.00 | 40.11 | 42.65 |
| *Open-Source Non-Reasoning Medical Finetuned Models* | | | | | | | | | | |
| MedGemma-27b-it | **49.61** | 44.20 | 48.09 | 43.45 | 41.36 | 46.58 | 50.33 | 45.62 | 39.70 | 45.47 |

### 3.5 DATA FILTERING

**Difficulty Filtering.** To ensure the high challenge of MEDFRAMEQA, we utilize 3 advanced MLLMs—GPT-4-Turbo-V (OpenAI, 2023b), o1 (Jaech et al., 2024), and GPT-4o (Hurst et al., 2024)—for further filtering. If *any* of the models selects the correct option, the question is deemed too easy and discarded. This step trims the pool from 4,457 to 3,654 items.

**Human Evaluation.** Additionally, we conduct a manual evaluation to eliminate entries featuring low-quality frames. In detail, we exclude entries with frames that are: (i) blurred or display overlapping visuals due to faulty video extraction; (ii) show recognizable human faces, infringing upon the privacy guidelines described in Section 3.2; (iii) devoid of significant visual medical content. As a result, 803 entries were excluded, yielding a final benchmark set of 2,851 high-quality entries.

## 4 EXPERIMENTS

### 4.1 DATA STATISTICS

In this section we summarize the data distribution of MEDFRAMEQA. Starting from the 3,420 instructional videos collected in Section 3.1, we extract 111,942 key-frames and retain 9,237 high-quality, medically relevant frames. These frames are used to construct 2,851 multi-image, closed-ended, single-choice VQA pairs, which span 9 human body systems and 43 organs, featuring 114 unique keyword combinations derived from the most common diseases and their associated diagnostic imaging modalities for each organ following Herring (2019). Each generated VQA pair consists of 2–5 frames, accompanied by a challenging question that requires integrating information across all provided frames to answer correctly. The composition of body systems, organs and modalities in MEDFRAMEQA is provided in Section B and shown in Figure 5 (a) (b) (c) respectively.

We stress that the defining feature of MEDFRAMEQA is that every question is tethered to multiple images, deliberately pushing models to reason across frames—a core requirement in real-world diagnosis. Concretely, among the 2,851 VQA items, 1,186 pairs contain 2 frames, 602 pairs contain 3 frames, 256 pairs contain 4 frames, and 807 pairs contain 5 frames. We also present the distribution of frames per question in Figure 5(e).

## 4.2 Models

We evaluate both proprietary and open-source MLLMs on MEDFRAMEQA, encompassing reasoning and non-reasoning models, with a particular focus on recent advancements in medical reasoning. For evaluation, we use the prompt template as in MMMU-pro(Yue et al., 2024b) (see Section F.4).

**Reasoning Models:** We evaluate MEDFRAMEQA on recent reasoning models, including the proprietary model `o4-mini` (OpenAI, 2025), `o3` (OpenAI, 2025), `o1` (Jaech et al., 2024), `Claude-3.7-Sonnet` (Anthropic, 2025) and `Gemini-2.5-Flash` (Google, 2025). We also include the open-source reasoning model `QvQ-72B-Preview` (Team, 2024).

**Non-Reasoning Models:** We also evaluate MEDFRAMEQA on non-reasoning models, including proprietary models, `GPT-4o` (Hurst et al., 2024), `GPT-4o-mini` (Hurst et al., 2024) and `GPT-4-Turbo-V` (OpenAI, 2023b). We also include the open-source model `Qwen2.5-VL-72B-Instruct` (Bai et al., 2025) the medical fine-tuned model `MedGemma-27b-it` (Sellergren et al., 2025) to evaluate domain-specific adaptations.

## 4.3 Main Results

**Advanced MLLMs struggle to holistically understanding multi-images.** Table 2 presents the evaluation of 11 advanced MLLMs on MEDFRAMEQA. In general, all assessed models demonstrate persistently low accuracy, with the peak accuracy remaining below 55.00%. To reduce model performance variability, for open-source models, we run each experiment three times and report the average results, whereas for proprietary models, we conduct only a single run due to API cost constraints. The proprietary model, `GPT-4o`, reaches an average accuracy of 45.67%, significantly lower in comparison to its performance on the single medical VQA benchmark (69.91% on VQA-RAD (Lau et al., 2018) as reported by Yan et al. (2024)). The leading open-source model, `Qwen2.5-VL-72B-Instruct`, achieves merely 42.65 ± 0.34% (SE) accuracy. To further verify that the suboptimal performance was attributable to deficient reasoning processes rather than inadequate medical knowledge, we evaluated `MedGemma-27b-it`, which similarly yielded poor results with 45.47 ± 0.59% (SE) accuracy. Together, these findings suggest that current advanced MLLMs fall short in capability to thoroughly analyze multiple medical images.

**Reasoning enhances multi-image understanding.** As shown in Table 2, we find that reasoning MLLMs consistently outperform non-reasoning ones. `Gemini-2.5-Flash` attains the highest accuracy among all models, notably outperforming the top non-reasoning model `GPT-4o` by 9.08% (54.75% *vs* 45.67%). Among the open-source models, `QvQ-72B-Preview` achieves an accuracy of 46.44% ± 0.66% (SE), showcasing a 3.79% enhancement compared to its non-reasoning counterpart, `Qwen2.5-VL-72B-Instruct`. This indicates that reasoning is particularly beneficial in clinical scenarios, which frequently involve multiple images.

**Overlooking or misinterpreting hinders reasoning across image sequence.** Despite the relatively enhanced performance of reasoning models, their performance is still limited. Our investigation reveals this arises from neglecting or misinterpreting the intermediary images during continuous reasoning over an image sequence. Here, we present a case study highlighting instances where `o1` fails to provide correct reasoning steps for questions in MEDFRAMEQA:

*Case 1: Neglegence of important information within multiple frames.* In Figure 3, we present a case where `o1` fails to integrate important information across multiple frames, leading to a flawed overall reasoning. While `o1` correctly identifies the "polar vessel sign" in the Doppler frame as suggestive of a parathyroid adenoma, it neglects anatomical cues from the transverse and sagittal views. These clearly demonstrate that the lesion is located posterior and inferior to the thyroid gland with a cranial-caudal orientation—hallmark features of parathyroid adenomas. By focusing solely on vascular features in a single frame and ignoring spatial localization in the others, `o1` misses the most definitive diagnostic evidence and produces an incorrect conclusion.

*Case 2: Mistake drawn from single image resulting in significant errors in subsequent reasoning.* In Figure 4, we present a case where `o1` fails to extract the correct spatial cue from a critical frame, leading to a cascade of reasoning errors. Specifically, `o1` misinterprets the axial image by identifying medial displacement of the nerve root, despite clear imaging evidence of lateral displacement caused by a foraminal disc herniation. This initial misjudgment propagates through

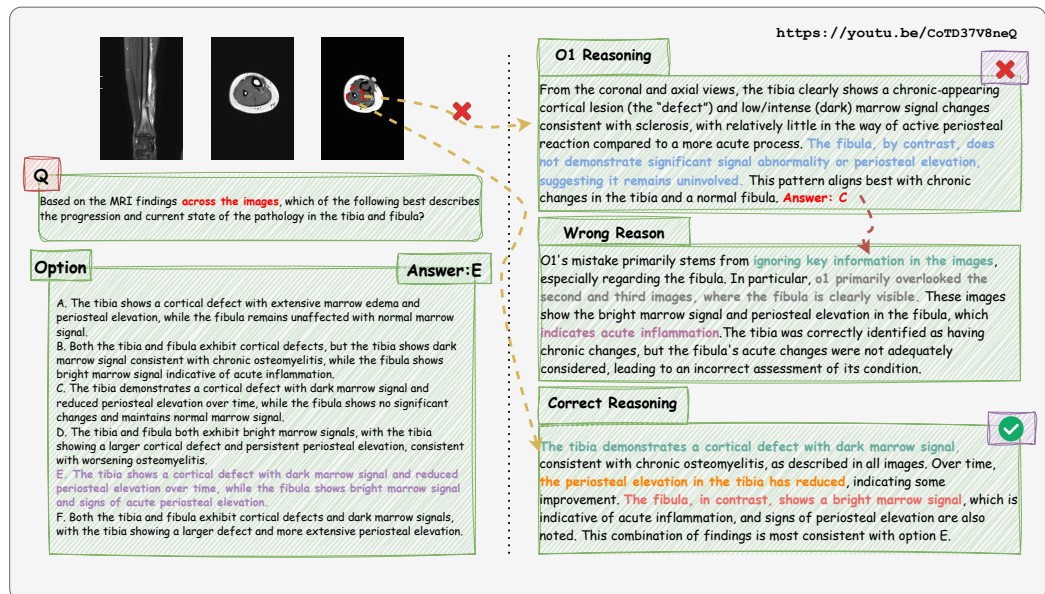

Figure 3: **Failure case study of o1 on MEDFRAMEQA.** Negligence of important information across multiple frames. In this case, o1 overlooked critical features in the second and third frames, which ultimately led to the selection of an incorrect answer.

Table 3: **Accuracy (%) of Models by Frame Count and Modality on MEDFRAMEQA.** We report the accuracy of models on questions in MEDFRAMEQA grouped by frame count with standard deviation (*SD*) and by modality. We empirically observe that accuracy fluctuates with increasing frame count and varies significantly across common imaging modalities.

| Model | Accuracy (%) by Frame Count | | | | | Accuracy (%) by Modality | | | | |
|---|---|---|---|---|---|---|---|---|---|---|
| | 2 | 3 | 4 | 5 | *SD* | CT | MRI | Ultrasound | X-ray | Other |
| o1 | 48.16 | 45.64 | 51.43 | 48.15 | 2.37 | 48.98 | 45.40 | 49.05 | 49.16 | 51.64 |
| o3 | 50.00 | 47.46 | 53.60 | 51.38 | 2.57 | 50.09 | 48.57 | 51.45 | 53.06 | 52.38 |
| o4-mini | 50.21 | 46.23 | 50.00 | 50.37 | 1.99 | 48.08 | 48.85 | 52.34 | 50.33 | 53.49 |
| Gemini-2.5-Flash | **53.54** | **55.48** | **55.47** | **55.76** | 1.02 | **54.57** | **53.60** | **57.36** | **58.14** | 49.24 |
| QvQ-72B-Preview | 48.00 | 46.73 | 42.32 | 45.23 | 2.12 | 45.18 | 47.62 | 48.32 | 44.08 | 47.98 |
| GPT-4-Turbo-V | 47.47 | 45.51 | 46.88 | 46.34 | 0.83 | 46.83 | 43.48 | 50.65 | 49.17 | 51.52 |
| GPT-4o | 47.30 | 45.18 | 40.23 | 45.35 | 3.01 | 45.52 | 43.27 | 48.58 | 47.51 | 51.52 |
| GPT-4o-mini | 35.16 | 36.21 | 32.42 | 33.09 | 1.77 | 35.26 | 34.31 | 34.88 | 34.55 | 29.55 |
| Claude-3.7-Sonnet | 49.41 | 48.01 | 51.56 | 50.68 | 1.55 | 50.75 | 49.11 | 49.10 | 49.83 | 46.21 |
| Qwen2.5-VL-72B-Instruct | 42.72 | 41.14 | 42.71 | 43.66 | 0.90 | 40.95 | 43.52 | 42.64 | 45.07 | 44.70 |
| MedGemma-27b-it | 43.73 | 44.80 | 46.88 | 48.08 | 1.70 | 47.64 | 43.03 | 44.10 | 43.19 | **54.08** |

its reasoning chain, ultimately resulting in the selection of an anatomically incorrect answer that contradicts the information integrated across both frames.

## 4.4 EVALUATION ACROSS ANATOMICAL STRUCTURES OR FRAME NUMBERS

**Comparisons between anatomical structures and modalities.** We report results for nine systems: Central Nervous System (**CNS**), Respiratory System (**RES**), Circulatory System (**CIR**), Digestive System (**DIG**), Urinary System (**URI**), Reproductive System (**REP**), Endocrine System (**END**), Musculoskeletal System (**MSK**), and Auxiliary (**AUX**). The system-wise performance we report in Table 2 reveals substantial variability in task difficulty. For instance, Gemini-2.5-Flash achieves an accuracy of 60.21% on questions related to the musculoskeletal system, but only 48.61% on the urinary system, resulting in an accuracy gap of 11.60 percentage points. In Section E, we present a detailed analysis of performance variation across four representative organs in MEDFRAMEQA. We also report the performance of MLLMs across different imaging modalities in Table 3. Notably, the accuracy varies significantly across common modalities such as CT, MRI, Ultrasound, and X-ray.

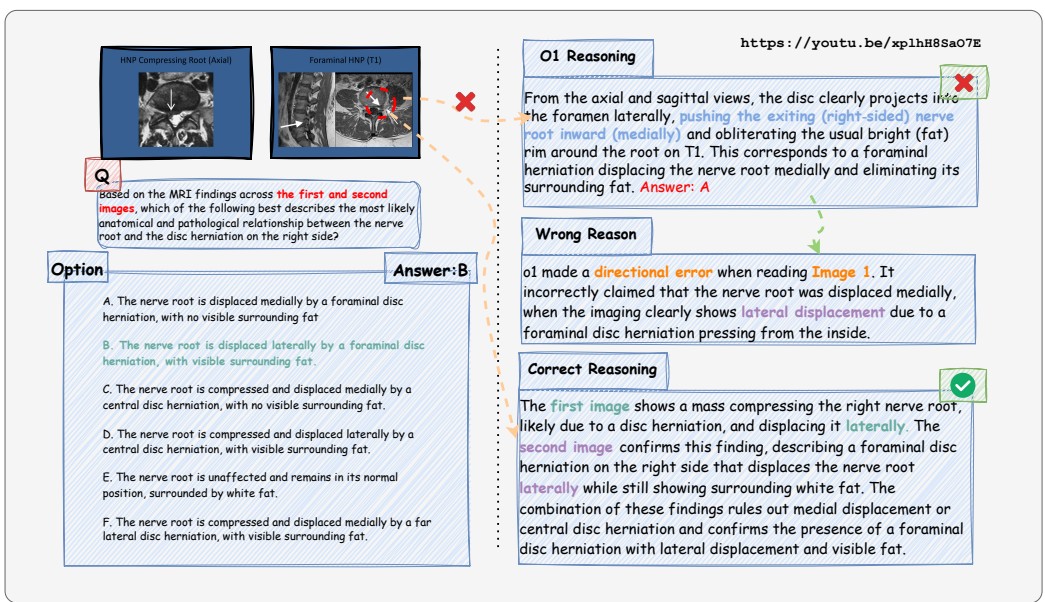

Figure 4: **Failure case study of o1 on MEDFRAMEQA.** A mistake originating from a single image can result in significant errors in subsequent reasoning. In this case, o1 made a directional error when interpreting the first frame, which propagated through its reasoning process and ultimately led to an incorrect answer.

QvQ-72B-Preview exhibits a 4.24% performance gap between Ultrasound and X-ray, whereas Gemini-2.5-Flash shows a 4.54% gap between MRI and X-ray.

These discrepancies across anatomical structures and modalities highlight the modality sensitivity of current MLLMs, suggesting that training should include more diverse and balanced modality-organ combinations to improve generalization.

**Comparisons betweem VQAs with different numbers of frames.** In Table 3, we report the accuracy of models on questions in MEDFRAMEQA, grouped by the number of frames each question contains. Empirically, we observe that accuracy fluctuates as the number of images per question increases, with performance improving at certain frame counts and declining at others. Among the MLLMs, GPT-4o exhibits substantial fluctuation, with a standard deviation of 3.01, whereas GPT-4-Turbo-V shows minimal variation, with a standard deviation of just 0.83. These fluctuations suggest that model performance is not strictly determined by the number of frames, but may instead be influenced by the complexity or redundancy of visual information across frames.

## 5  CONCLUSION AND LIMITATIONS

This paper introduces MEDFRAMEQA, a multi-image medical visual question answering benchmark, comprising 2851 multi-image multi-choice questions, sourced from 3420 medical videos of 114 keywords and covering over 43 organs. We also propose an automated pipeline to generate high-quality multi-image VQA data from YouTube while ensuring semantic progression and contextual consistency across frames. Unlike existing datasets that rely on single-image inputs or lack detailed reasoning about the answer, MEDFRAMEQA has both multi-image question answering pairs and a detailed reasoning process, containing 2-5 images input and 3.24 images input per question. We comprehensively benchmark ten state-of-the-art models, presenting accuracies predominantly below 50%. While MEDFRAMEQA reveals clear evidence of current MLLMs' inability in handling multi-image questions of clinical reasoning, effective strategies to enhance their multi-image reasoning capabilities remain underexplored. Future work will focus on developing and evaluating methods to improve such capabilities. We believe MEDFRAMEQA will serve as a valuable resource for advancing research in multimodal medical AI and fostering the development of more capable diagnostic reasoning systems.

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

## A USE OF LLMS

We employed large language models (LLMs) in the dataset construction pipeline to refine and filter captions, identify and merge semantically related captions, and generate multi-image VQA items. We further benchmarked state-of-the-art MLLMs on MedFrameQA.

During the preparation of this manuscript, we used OpenAI's GPT-4.1 model for minor language refinement and smoothing of the writing. The AI tool was not used for generating original content, conducting data analysis, or formulating core scientific ideas. All conceptual development, experimentation, and interpretation were conducted independently without reliance on AI tools.

## B DATA DISTRIBUTION

We present detailed data distributions across body systems, organs, and imaging modalities in Figure 5(a), (b), and (c), respectively. A word cloud of keywords in MEDFRAMEQA is shown in Figure 5(d), and the distribution of frame counts per question is provided in Figure 5(e).

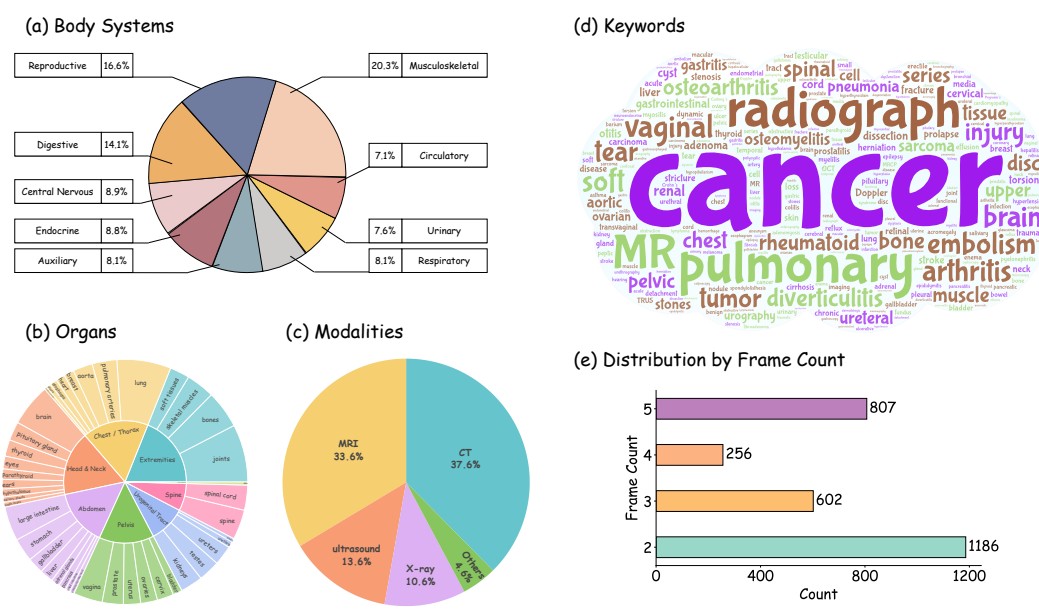

Figure 5: **Data distribution of MEDFRAMEQA.** In Figure 5(a), we show the distribution across body systems; (b) presents the distribution across organs; (c) shows the distribution across imaging modalities; (d) provides a word cloud of keywords in MEDFRAMEQA; and (e) reports the distribution of frame counts per question.

## C API COST

Generation of each data entry costs 5 times calling of `GPT-4o` API on average, depending on the number of frames involved in the data entry. Construction of 2,851 data entries costs 14,255 API calls in total.

For proprietary models (e.g., `GPT-4o`, `Gemini-2.5-Flash`, `Claude-3.7-Sonnet`), we use their official APIs and perform 2,851 requests per model, corresponding to the number of examples in MEDFRAMEQA.

For open-source models (e.g., `QvQ-72B-Preview`, `Qwen2.5-VL-72B-Instruct`, `MedGemma-27b-it`), we conducted three independent runs on 4×A100 GPUs and calculated error bars. Due to API quota constraints, proprietary models were evaluated only once.

# D  KEYWORD LIST

| System | Organ | Keyword |
|---|---|---|
| central nervous system | brain | stroke CT |
| | | brain tumor MRI |
| | | cerebral hemorrhage CT |
| | | epilepsy EEG imaging |
| | | traumatic brain injury CT |
| | spinal cord | spinal cord injury MRI |
| | | disc herniation MRI |
| | | spinal stenosis CT |
| | | myelitis MRI |
| respiratory system | lung | pneumonia chest radiograph |
| | | lung cancer CT |
| | | pulmonary embolism CT angiography |
| | | chronic obstructive pulmonary disease CT |
| | trachea bronchi | bronchial asthma bronchography |
| | pleura | pleural effusion ultrasound |
| circulatory system | heart | coronary artery disease angiography |
| | | heart failure echocardiography |
| | | myocardial infarction CT |
| | | cardiomyopathy MRI |
| | pulmonary arteries | pulmonary embolism CT angiography |
| | | pulmonary hypertension CT |
| | aorta | aortic aneurysm CT |
| | | aortic dissection MR angiography |
| digestive system | esophagus | esophageal cancer CT |
| | | gastroesophageal reflux esophagram |
| | | esophageal stricture endoscopic ultrasound |
| | stomach | gastric cancer CT |
| | | peptic ulcer gastroscopy |
| | | gastritis upper gastrointestinal series |
| | liver | liver cirrhosis CT |
| | | hepatocellular carcinoma MRI |
| | | hepatitis ultrasound |
| | pancreas | pancreatic cancer CT |
| | | acute pancreatitis CT |
| | | chronic pancreatitis MRCP |
| | gallbladder | gallstones ultrasound |
| | | cholecystitis HIDA scan |
| | | gallbladder cancer CT |
| | small intestine | Crohn's disease MRI enterography |
| | | small bowel obstruction CT |
| | | intestinal bleeding capsule endoscopy |
| | large intestine | colorectal cancer colonoscopy |
| | | diverticulitis CT |
| | | ulcerative colitis barium enema |
| urinary system | kidneys | kidney stones CT |
| | | renal cell carcinoma MRI |
| | | pyelonephritis ultrasound |
| | ureters | ureteral stones CT urography |
| | | ureteral stricture MR urography |
| | bladder | bladder cancer cystoscopy |
| | | urinary tract infection ultrasound |
| | | bladder stones CT |
| | urethra | urethral stricture urethrography |
| | | urethral injury CT urethrography |
| reproductive system | testes | testicular cancer ultrasound |
| | | testicular torsion Doppler ultrasound |
| | | epididymitis ultrasound |
| | prostate | prostate cancer MRI |
| | | benign prostatic hyperplasia TRUS |
| | | prostatitis pelvic CT |

| System | Organ | Keyword |
|---|---|---|
| | penis | erectile dysfunction Doppler ultrasound
Peyronie's disease MRI |
| | ovaries | ovarian cyst ultrasound
ovarian cancer MRI
polycystic ovary syndrome ultrasound |
| | uterus | endometrial cancer MRI
uterine fibroids ultrasound
adenomyosis pelvic MRI |
| | cervix | cervical cancer MRI
cervical dysplasia colposcopy |
| | vagina | vaginal cancer MRI
vaginal prolapse transvaginal ultrasound |
| endocrine system | thyroid | thyroid nodule ultrasound
thyroid cancer scintigraphy
hyperthyroidism neck CT |
| | parathyroid | parathyroid adenoma scintigraphy
hyperparathyroidism ultrasound |
| | adrenal glands | adrenal adenoma CT
pheochromocytoma MRI
Cushing's syndrome adrenal scintigraphy |
| | pancreas (endocrine) | insulinoma CT
pancreatic neuroendocrine tumor MRI |
| | pituitary gland | pituitary adenoma MRI
acromegaly dynamic MRI |
| | hypothalamus | hypothalamic tumor MRI
hypopituitarism functional MRI |
| musculoskeletal system | bones | osteoporosis DEXA
bone fracture radiograph
osteomyelitis MRI |
| | joints | osteoarthritis radiograph
rheumatoid arthritis MRI
joint effusion ultrasound |
| | skeletal muscles | muscle tear MRI
myositis ultrasound
muscular dystrophy EMG imaging |
| | spine | disc herniation MRI
spinal stenosis CT
spondylolisthesis radiograph |
| auxiliary systems and tissues | eyes | glaucoma OCT
retinal detachment ultrasound
macular degeneration fundus photography |
| | ears | otitis media temporal bone CT
hearing loss brain MRI |
| | skin | melanoma confocal microscopy
skin cancer dermatologic ultrasound |
| | lymph nodes | lymphoma CT
lymphadenitis ultrasound |
| | soft tissues | soft tissue sarcoma MRI
lipoma ultrasound |
| | salivary glands | salivary gland tumor ultrasound
sialadenitis sialography |
| | breast | breast cancer mammography
fibroadenoma ultrasound
breast cyst MRI |

# E  COMPARISON OF ORGANS

We present a detailed organ-wise accuracy comparison of ten state-of-the-art MLLMs on MED-FRAMEQA. Our results reveal substantial performance variation across different organs. While `Gemini-2.5-Flash` outperforms other models on average in Table 2, open-source models like `QvQ-72B-Preview` demonstrate competitive performance on specific organs, such as the ureters and pulmonary arteries. This variability highlights the sensitivity of MLLM performance to the anatomical structures involved, underscoring the need to develop models that are more robust to anatomical diversity. This variability underscores the sensitivity of MLLM performance to organ-specific features and highlights the need for future research focused on improving anatomical generalization across a wide range of clinical scenarios.

Table 4: **Accuracy of Models by organs on MEDFRAMEQA.** We report the organ-wise accuracy of the models on MEDFRAMEQA. The best accuracy is highlighted in bold.

| Organs | Model Accuracy | | | | | | | | | | |
|---|---|---|---|---|---|---|---|---|---|---|---|
| | Gemini-2.5-Flash | Claude-3.7-Sonnet | o4-mini | o3 | o1 | GPT-4o | GPT-4o-mini | GPT-4-Turbo-V | QvQ-72B | Qwen2.5-VL-72B-Instruct | MedGemma-27b-it |
| **auxiliary systems and tissues** | | | | | | | | | | | |
| soft tissues | **48.65** | 37.84 | 45.95 | 39.19 | 35.14 | 36.49 | 32.43 | 35.14 | 40.54 | 30.63 | 35.68 |
| salivary glands | 55.00 | 50.00 | 45.00 | 52.63 | 47.37 | 40.00 | 40.00 | 45.00 | **66.67** | 48.33 | 43.33 |
| skin | 33.33 | 66.67 | 50.00 | 70.00 | 54.55 | **75.00** | 41.67 | **75.00** | 36.11 | 63.89 | 50.00 |
| breast | 52.63 | 55.26 | 55.26 | 57.89 | **58.33** | 42.11 | 39.47 | 39.47 | 50.88 | 35.09 | 41.23 |
| lymph nodes | 61.11 | **77.78** | 72.22 | 72.22 | 61.11 | 55.56 | 27.78 | 61.11 | 53.70 | 55.56 | 53.70 |
| ears | **58.33** | 47.22 | 44.44 | 52.78 | 57.14 | 50.00 | 30.56 | 55.56 | 46.30 | 37.04 | 40.74 |
| eyes | **56.25** | 50.00 | 54.17 | 46.81 | 51.06 | 43.75 | 37.50 | 52.08 | 47.22 | 45.83 | 36.11 |
| **central nervous system** | | | | | | | | | | | |
| brain | 50.00 | 49.38 | 42.41 | 45.86 | 46.05 | 51.25 | 44.38 | 46.88 | 42.92 | 42.50 | **51.87** |
| spinal cord | 46.81 | 48.94 | **52.13** | 51.06 | 48.35 | 44.68 | 37.23 | 42.55 | 48.23 | 44.33 | 45.74 |
| **circulatory system** | | | | | | | | | | | |
| pulmonary arteries | 54.84 | **56.99** | 50.54 | 49.46 | 51.09 | 43.01 | 44.09 | 47.31 | 51.97 | 44.09 | 49.82 |
| aorta | **60.81** | 48.65 | 45.21 | 50.00 | 45.83 | 35.14 | 35.14 | 41.89 | 43.69 | 40.09 | 52.70 |
| heart | **55.88** | 52.94 | 51.52 | 51.52 | 53.12 | 26.47 | 35.29 | 32.35 | 43.14 | 42.16 | 37.25 |
| **digestive system** | | | | | | | | | | | |
| large intestine | 47.29 | 47.29 | 42.64 | 38.28 | 41.73 | **48.06** | 23.26 | 46.51 | 35.14 | 31.52 | 37.98 |
| esophagus | 59.26 | 51.85 | **70.37** | 62.96 | 59.26 | 62.96 | 22.22 | 62.96 | 61.73 | 38.27 | 60.49 |
| small intestine | 61.11 | 55.56 | **72.22** | 58.82 | 62.50 | 44.44 | 16.67 | 55.56 | 46.30 | 50.00 | 55.56 |
| gallbladder | 37.70 | 44.26 | 34.43 | 38.33 | 41.38 | 40.98 | 39.34 | **47.54** | 40.98 | 36.61 | 39.34 |
| stomach | 59.09 | 59.09 | 55.17 | **60.00** | 54.12 | 57.95 | 32.95 | 56.82 | 37.88 | 51.14 | 46.59 |
| liver | 54.90 | 54.90 | 52.94 | **60.78** | 52.94 | 50.98 | 29.41 | 43.14 | 54.25 | 46.41 | 43.14 |
| pancreas | 39.29 | 35.71 | 42.86 | 39.29 | 35.71 | 42.86 | 25.00 | 42.86 | 32.14 | 32.14 | **44.05** |
| **endocrine system** | | | | | | | | | | | |
| pancreas (endocrine) | 41.18 | 35.29 | **52.94** | 35.29 | 35.29 | 41.18 | 17.65 | 41.18 | 35.29 | 25.49 | 29.41 |
| hypothalamus | **56.67** | 43.33 | 53.85 | 50.00 | 42.31 | 46.67 | 43.33 | 46.67 | 45.56 | 45.56 | 52.22 |
| parathyroid | 56.41 | 38.46 | 47.37 | 50.00 | 57.14 | 41.03 | 35.90 | 46.15 | 49.57 | 47.86 | **60.68** |
| pituitary gland | 56.34 | 56.34 | **59.15** | 57.75 | 56.52 | 45.07 | 21.13 | 47.89 | 57.28 | 52.11 | 54.93 |
| adrenal glands | **53.12** | 43.75 | **53.12** | 43.75 | 25.00 | **53.12** | 40.62 | 43.75 | 41.67 | 27.08 | 45.83 |
| thyroid | 58.06 | 51.61 | 46.77 | 55.74 | 50.00 | 48.39 | 30.65 | **61.29** | 43.01 | 41.40 | 45.70 |
| **musculoskeletal system** | | | | | | | | | | | |
| spine | 57.14 | 49.11 | 48.21 | **58.04** | 48.65 | 47.32 | 35.71 | 50.00 | 48.81 | 46.43 | 48.51 |
| bones | **62.68** | 50.70 | 51.77 | 56.83 | 54.07 | 43.66 | 37.32 | 38.03 | 55.16 | 40.38 | 41.31 |
| skeletal muscles | **63.55** | 61.68 | 62.62 | 54.29 | 50.94 | 45.79 | 38.32 | 51.40 | 50.78 | 56.39 | 57.63 |
| joints | **58.53** | 50.69 | 52.53 | 52.31 | 51.87 | 40.55 | 31.34 | 44.24 | 45.16 | 39.02 | 41.01 |
| **reproductive system** | | | | | | | | | | | |
| vagina | **56.88** | 50.46 | 44.44 | 47.17 | 38.24 | 49.54 | 35.78 | 54.13 | 48.01 | 43.12 | 52.60 |
| penis | 42.86 | 28.57 | 28.57 | 14.29 | 14.29 | 42.86 | 28.57 | 50.00 | 38.10 | **52.38** | 45.24 |
| ovaries | 50.79 | 47.62 | 44.44 | 46.77 | 52.54 | 42.86 | 22.22 | 38.10 | 49.74 | **55.03** | 47.62 |
| prostate | **50.63** | 49.37 | 40.51 | 42.86 | 30.26 | 46.84 | 43.04 | 48.10 | 40.93 | 39.66 | 45.57 |
| cervix | **61.29** | 53.23 | 41.67 | 38.98 | 47.37 | 48.39 | 32.26 | 48.39 | 44.09 | 40.32 | 40.32 |
| testes | **64.20** | 46.91 | 46.91 | 51.25 | 52.50 | 44.44 | 34.57 | 45.68 | 54.73 | 43.21 | 44.44 |
| uterus | 52.31 | 40.00 | 46.15 | 46.88 | 42.19 | 41.54 | 32.31 | **53.85** | 45.13 | 38.46 | 45.64 |
| **respiratory system** | | | | | | | | | | | |
| trachea bronchi | 50.00 | 60.00 | 55.56 | 62.50 | 55.56 | 70.00 | 30.00 | 50.00 | 46.67 | **73.33** | 66.67 |
| lung | **59.11** | 47.29 | 50.25 | 53.00 | 50.51 | 48.28 | 35.96 | 45.32 | 47.62 | 46.96 | 43.68 |
| pleura | **52.94** | 23.53 | 41.18 | 35.29 | 25.00 | 47.06 | 47.06 | **52.94** | 35.29 | 37.25 | 37.25 |
| **urinary system** | | | | | | | | | | | |
| ureters | 44.59 | 44.59 | 40.54 | 46.48 | 42.65 | 40.54 | 25.68 | 45.95 | 41.89 | 37.84 | **48.20** |
| kidneys | 50.00 | 51.19 | **58.33** | 50.00 | 54.32 | 50.00 | 38.10 | 46.43 | 44.84 | 40.48 | 34.52 |
| urethra | 52.17 | 43.48 | **60.87** | 43.48 | 40.91 | 21.74 | 47.83 | 26.09 | 52.17 | 49.28 | 36.23 |
| bladder | 51.43 | 57.14 | 54.29 | **65.71** | 54.29 | 51.43 | 42.86 | 40.00 | 51.43 | 36.19 | 46.67 |

# F PROMPT DETAILS

## F.1 FILTER AND REPHRASE CAPTIONS

**Prompts for Filtering Non-Medical Image and Rephrasing**

**prompt_template:**

You are given the following:
- A **keyframe image** extracted from a YouTube video retrieved using the keyword "{keyword}", which relates to the "{organ}" in the "{system}".
- The keyframe corresponds to the time interval: [{frame_start_time}, {frame_end_time}] in the video. You may assume that visual content remains stable during this period.
- A **list of caption segments**, spanning from {start_time} to {end_time} seconds, provided as a JSON array in the `{caption_json_list}` variable. These segments represent the spoken content near the frame's timestamp and may contain information that helps describe or interpret the keyframe image. Each caption object contains:
- `"startTime"`: start time in seconds
- `"endTime"`: end time in seconds
- `"sentence"`: caption content

### Your Task

1. **Determine Benchmark Eligibility**:
Answer these questions to guide your reasoning:
1. Does the image prominently depict clear, authentic medical imaging relevant to "{keyword}" (e.g., sharp radiographs or scans, including multiple images if they are all visible and relevant)?
2. Is the image **primarily composed of medical imaging**, even if there are text overlays or minor visual obstructions?
3. Is the image suitable for inclusion in a medical benchmark dataset (e.g., sharp, intelligible, and relevant to medical imaging, with at least 85% of the image area consisting of meaningful medical imaging, excluding blank regions, borders, or irrelevant content)?
4. Is the image free of any unrelated human faces, including but not limited to presenters in video conference screenshots (e.g., Zoom speaker windows) or other non-medical human portraits?

2. **Faithful Rephrasing**:
- Rephrase the caption into a coherent, fluent, and high-quality medical description of the visual content of the current frame, as conveyed solely by the dialogue in the provided captions.
- The description must use precise medical terminology and reflect a medical imaging context (e.g., radiology or anatomy).
- **Include only information explicitly stated in the captions that directly relates to the current frame's visual content**, such as descriptions, identifications, observations, questions, answers, corrections, and transitional statements.
- Strictly avoid any details not present in the captions, including information from the image itself, external context, or unrelated dialogue (e.g., discussions about other frames or topics).

### Output Format

Return your answer as a valid JSON object, you **should not include markdown in your output**:
{{
    "result": "yes" | "no",
    "reason": "A concise explanation (max 50 words) for why the image is or is not suitable for the benchmark.",
    "captions": all the captions combined together,
    "rephrased_description": "A faithful and fluent rephrasing of the caption content, without hallucination."
}}

If the image is **not** suitable for the benchmark (i.e., `"result": "no"`), then only return the following fields in your output, you **should not include markdown in your output**:
{{
    "result": "no",
    "reason": "A concise explanation (max 50 words) for why the image is not suitable for the benchmark.",
}}

## F.2 TRANSCRIPTS RELATION CHECK

---

**Prompts for Pairing Related Captions**

**prompt_template:**
You are given one or more caption segments corresponding to one or more continuous medical keyframes from a video. You do not have access to the actual images.

These caption segments come from a medical video retrieved using the keyword "{keyword}", and are related to the body part "{body_part}". Each caption describes the anatomical structures or procedural content visible in its corresponding keyframe.

Your task is to analyze the content of all caption segments and determine which segments are discussing **the same or closely related medical topic or structure** (e.g., same procedure, same organ, or same pathology). Group together all captions that appear to describe the same medical subject. Each group should represent a coherent topic or issue that could be visually identifiable in the corresponding keyframes.
Below are all the caption segments:
```

{caption}
```

**Requirements:**
- Focus only on medically or visually coherent topics.
- Do not group captions based only on linguistic similarity—there must be a medically meaningful connection.
- Each group must contain at least one caption.
- If a caption clearly describes a different topic from others, place it in its own group.
- For each group, provide a brief explanation in the reason field describing why these captions are grouped together.

**Output Format:**
The output must strictly follow the JSON format below (no markdown, no explanations):
```
{{
    "frames": [all the caption numbers],
    "pairs_of_related_frames": [
    {{
        "selected_captions": [1, 2],
        "related_reason": "Both captions describe the insertion of a catheter into the same artery."
    }},
    {{
        "selected_captions": [3],
        "related_reason": "This caption describes a different procedure involving the venous system."
    }}
    ]
}}
```

## F.3 MULTI-FRAME VQA PAIR GENERATION

---

### Prompts for Generating VQA pairs

**prompt_template:**
Your task is to generate **expert-level, medically valuable** question that:

- Uses **every piece of visual information contained in the captions** (treat the captions only as your private description of each image).
- Demands advanced competencies such as anatomical reasoning, differential diagnosis, pathology identification, or procedural planning.
- Is grounded **solely in what can be seen on the images.** Do not add outside facts unless the finding is directly evident from the described appearance.
- Refers to each picture as "first image", "second image", etc. in the order implied by the captions.
- Never hints at, quotes, or mentions the captions, videos, or any textual description. All wording must make it seem as though the questioner has the images in front of them.
- Add as many plausible but misleading distractors as possible (commonly 4–6 or more). Craft the incorrect answer choices so they are commonly confused with the correct diagnosis/procedure given the depicted findings, thereby maximizing the likelihood of error for anyone who has not carefully interpreted every visual detail.
- Important: Do not generate questions that test theoretical definitions, textbook knowledge, or general medical concepts alone. Only generate questions whose answers depend on observing specific visual features explicitly described in the captions. Do not ask about general patterns like 'penumbra parameters'—instead, ask how those parameters appear in the actual image described.

**Below are all the caption segments:**

```
{caption}
```

**Output Format (strict JSON structure, no markdown allowed):**
```
{{
  "related_captions": ["caption_1", "caption_2", ...],
  "mcq_questions": [
  {{
    "question": "A medically grounded visual question requiring comparison across the provided images.",
    "options": ["Option A", "Option B", "Option C", "Option D", ...],
    "correct_answer": "Please select the best answer from the given options.",
    "reasoning_chain": "A clear explanation of how the correct answer is visually derived by integrating details
      from all related images.",
    "supporting_segments": {{
    "caption_1": "Supporting phrase from caption_1.",
    "caption_2": "Supporting phrase from caption_2.",
    "...": "Add additional quotes as needed."
    }}
  }}
  ]
}}
```

## F.4 BENCHMARK EVALUATION

---

**Prompts for Evaluation**

**GPT Series & Claude & Qwen Series Prompt Template:**

Answer the following multiple-choice question. Images are provided. The last line of your response should be strictly of the following format: 'Answer: $LETTER' (without quotes) where LETTER is one of the options. For example, if the correct answer is A, your response should be: 'Answer: A'. Think step by step before answering.
Question:{question}

Options:
{options}

------------------------------------------------------------------------------------

**Gemini Prompt Template:**

Answer the following multiple-choice question. Images are provided. The last line of your response should be strictly of the following format: 'The final answer is $\\boxed{{LETTER}}$' (without quotes) where LETTER is one of the options. For example, if the correct answer is A, your response should be: 'The final answer is $\\boxed{{A}}$'. Think step by step before answering.
Question:{question}

Options:
{options}

------------------------------------------------------------------------------------

**QVQ Prompt Template:**

Answer the following multiple-choice question. Images are provided. The last line of your response should be strictly of the following format: '**Final Answer**\n\n\\[ \\boxed{{LETTER}} \\]' (without quotes) where LETTER is one of the options. For example, if the correct answer is A, your response should be: '**Final Answer**\n\n\\[ \\boxed{{A}} \\]'. Think step by step before answering.
Question:{question}

Options:
{options}

## G REPRESENTATIVE EXAMPLES

### G.1 TWO FRAMES EXAMPLE

**Example - #1**

**System**: respiratory system
**Organ**: lung
**Modality**: X-ray
**YouTube Link**: https://youtu.be/J1n2mJ00xKs

. . . . . . . . . . . . . . . . . . . . . . . . . . . . . . . . . . . . . . . . . . . . . . . . . . . . . . . . . . . . . . .

**Input Images:**

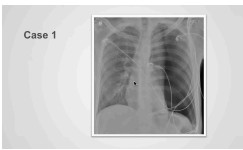 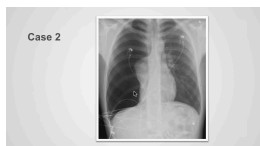

**Question:**
Based on the radiographic findings in the first and second images, which of the following best describes the side and severity of the pneumothorax, as well as the associated mediastinal shift?

**Options:**
A. A large left-sided pneumothorax with mediastinal shift to the right, as seen in the first image, and a large right-sided pneumothorax with mediastinal shift to the left, as seen in the second image.
B. A small left-sided pneumothorax with no mediastinal shift in the first image, and a large right-sided pneumothorax with mediastinal shift to the left in the second image.
C. A large right-sided pneumothorax with mediastinal shift to the left in both the first and second images.
D. A large left-sided pneumothorax with no mediastinal shift in the first image, and a large right-sided pneumothorax with mediastinal shift to the right in the second image.
E. A large left-sided pneumothorax with mediastinal shift to the left in the first image, and a large right-sided pneumothorax with mediastinal shift to the right in the second image.
F. A small right-sided pneumothorax with no mediastinal shift in the first image, and a large left-sided pneumothorax with mediastinal shift to the right in the second image.

. . . . . . . . . . . . . . . . . . . . . . . . . . . . . . . . . . . . . . . . . . . . . . . . . . . . . . . . . . . . . . .

**Answer: A**

**Reason:**
The first image shows a large left-sided pneumothorax, evidenced by the additional line between the third and fourth ribs and the collapse of the left lung, with mediastinal structures displaced to the right. The second image depicts a large right-sided pneumothorax, as indicated by the significant collapse of the right lung and mediastinal shift to the left. These findings are consistent with the descriptions provided in both images.

## G.2 THREE FRAMES EXAMPLE

### Example - #2

**System**: central nervous system
**Organ**: brain
**Modality**: CT
**YouTube Link**: https://youtu.be/eoxKSAoGW2s

. . . . . . . . . . . . . . . . . . . . . . . . . . . . . . . . . . . . . . . . . . . . . . . . . . . . . . . . . . . .

**Input Images:**

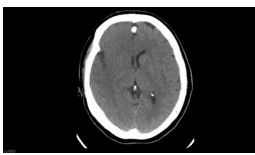 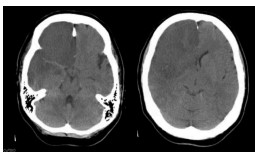 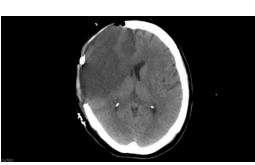

**Question:**
Based on the imaging findings across the first, second, and third images, which of the following best describes the most likely etiology and current stage of the observed pathology?

**Options:**
A. Acute ischemic stroke involving the right anterior cerebral artery and middle cerebral artery territories with significant mass effect and early signs of herniation.
B. Chronic ischemic changes in the right cerebral hemisphere with resolved mass effect and no evidence of acute intervention.
C. Acute hemorrhagic stroke in the right cerebral hemisphere with associated midline shift and ventricular compression.
D. Acute ischemic stroke involving the left anterior cerebral artery and middle cerebral artery territories with no significant mass effect.
E. Subacute ischemic stroke in the right cerebral hemisphere with complete resolution of midline shift following surgical intervention.
F. Acute ischemic stroke involving the right anterior cerebral artery and middle cerebral artery territories with persistent mass effect despite surgical decompression.

. . . . . . . . . . . . . . . . . . . . . . . . . . . . . . . . . . . . . . . . . . . . . . . . . . . . . . . . . . . .

**Answer: A**

**Reason:**
The correct answer is derived by integrating the following visual findings: (1) The first image shows diffuse parenchymal hypodensity, sulcal effacement, and poor gray-white differentiation in the right hemisphere, consistent with acute ischemic stroke. (2) The second image highlights a well-defined area of low attenuation involving the anterior and middle cerebral artery territories, with significant mass effect, ventricular compression, and midline shift, further supporting acute ischemic stroke with complications. (3) The third image confirms the persistence of poor gray-white differentiation and mass effect, despite surgical decompression, indicating an acute stage of ischemic stroke with ongoing complications. The other options are incorrect because they either misidentify the hemisphere, mischaracterize the pathology as hemorrhagic, or incorrectly describe the stage or resolution of the findings.

## G.3 FOUR FRAMES EXAMPLE

### Example - #3

**System**: musculoskeletal system
**Organ**: spine
**Modality**: X-ray
**YouTube Link**: https://youtu.be/U57_gD8Ftkw

.............................................................................

**Input Images:**

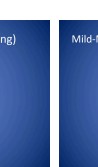 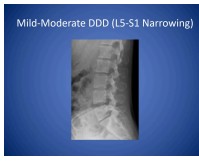 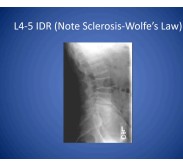 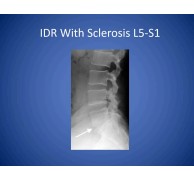

**Question:**
Based on the radiographic findings across the described images, which combination of features most strongly supports the diagnosis of advanced intervertebral disc degeneration at L4-5 compared to L5-S1?

**Options:**
A. Significant disc narrowing at L4-5 to 3-4 mm, sclerosis of the end plates, and absence of a vacuum phenomenon.
B. Disc narrowing at L4-5 to 3-4 mm, intact end plates without sclerosis, and whitening of the end plates due to prolonged stress.
C. Disc narrowing at L4-5 to 3-4 mm, sclerosis of the end plates, whitening of the end plates, and absence of fluid leading to a vacuum phenomenon.
D. Thick disc at L5-S1 measuring 12-13 mm, intact end plates without sclerosis, and absence of a vacuum phenomenon.
E. Disc narrowing at L4-5 to 3-4 mm, intact end plates without sclerosis, and thick disc at L5-S1 measuring 12-13 mm.
F. Disc narrowing at L4-5 to 3-4 mm, sclerosis of the end plates, whitening of the end plates, and thick disc at L5-S1 measuring 12-13 mm.

.............................................................................

**Answer: C**

**Reason:**
The correct answer integrates multiple visual findings described across the images. The significant narrowing of the L4-5 disc space to 3-4 mm is a hallmark of advanced degeneration, as noted in all images. Sclerosis of the end plates at L4-5 is explicitly described in image_2 and image_3, indicating chronic stress. Whitening of the end plates due to prolonged stress is mentioned in image_3 and image_4, further supporting degeneration. The vacuum phenomenon, described in image_3 and image_4, is a key feature of advanced degeneration, where gas accumulates in the disc space due to fluid loss. These findings collectively point to advanced intervertebral disc degeneration at L4-5 compared to L5-S1, which remains thicker and less degenerated.

## G.4 FIVE FRAMES EXAMPLE

---

### Example - #4

**System**: respiratory system
**Organ**: lung
**Modality**: CT
**YouTube Link**: https://youtu.be/rmX9AeGYscU

.....................................................................

**Input Images:**

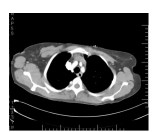 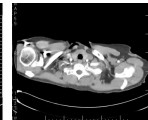 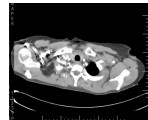 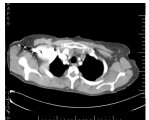 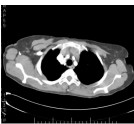

**Question**:
In the provided CT angiography images of the chest, which anatomical structure is most likely to serve as the primary landmark for orienting the scan and differentiating between the mediastinal and pulmonary vasculature regions?

**Options:**
A. Aortic arch
B. Left atrium
C. Right pulmonary artery
D. Superior vena cava
E. Descending thoracic aorta
F. Main pulmonary artery

.....................................................................

**Answer: A**

**Reason:**
The aortic arch is explicitly described across all images as the key landmark for orientation in the CT angiography scans. It is a readily identifiable structure that helps in distinguishing the mediastinal anatomy from the pulmonary vasculature. Other options, such as the left atrium or right pulmonary artery, are part of the chest anatomy but are not emphasized as primary orientation landmarks in the described images.

---

