# OpenReview forum: "MedFrameQA: A Multi-Image Medical VQA Benchmark for Clinical Reasoning"
_ICLR.cc/2026/Conference — ICLR 2026 Conference Withdrawn Submission_

### Official Review · Reviewer_vTBj · 2025-10-21

**Soundness:** 1
**Presentation:** 2
**Contribution:** 2
**Rating:** 2
**Confidence:** 4

**Summary:**

This paper proposes a benchmark for assessing the clinical reasoning capabilities of medical multimodal LLMs on multi-image visual question answering problems. To build the dataset, authors extract medical image sequences and corresponding captions from educational videos, and construct VQA problems by leveraging LLMs.Evaluation of state-of-the-art proprietary and open-source models on the proposed benchmark highlights their weakness of combining information across multiple medical images.

**Strengths:**

- The paper is clearly motivated: in medical practice, experts often look at multiple images of various imaging modalities and draw conclusions based on information integrated across multiple views, a key capability so far overlooked in single-image benchmarks.

- The paper calls attention to the shortcomings of current models in leveraging multiple images in medical decision making, highlighting important directions for future work.

**Weaknesses:**

- The dataset synthesis pipeline desperately needs expert evaluation. Currently, GPT-4o is used to judge the quality and utility of extracted frames, and to correct and combine captions across multiple views. While this is an acceptable strategy to generate candidate VQAs, medical professionals are necessary to review such candidates, and correct mistakes. Otherwise, we are relying on a dataset generated by models with the same shortcomings that we are attempting to evaluate. The only step where human filtering is involved is only focused on low-level visual quality (blurring) and detection of human faces, but does not check for sensibility from a medical perspective. Thus, the quality and reliability of the benchmark is questionable.

- Human performance is not evaluated on the benchmark, thus it is unclear how difficult the task is (there is no expert involved in the dataset curation loop as highlighted above). Is the benchmark even solvable by medical professionals?

- It is not ensured in any way that the models need to integrate information across multiple images. This could be demonstrated by dropping images from the VQAs one-by-one and demonstrating that performance drops to random guessing levels (that is other images are necessary in answering correctly).

- The difficulty filtering methodology is unclear/unreasonable to me. Questions are filtered out if *any* of 3 strong MLLMs can correctly answer them. If this filtering strategy is used how come the reported accuracies for these models is not 0? This filtering seems too aggressive.

- Basic characteristics of the benchmark, such as number of answer options, is missing from the paper and has to be inferred from plots.

- It is unclear to me how the temporal dimension of image frames is incorporated into the benchmark. Do we assume that the frames describe the state of the patient at the moment of diagnosis? Are they describing the patient's medical history? Is this information incorporated into the VQA problem?

**Questions:**

- How can authors ensure the correctness, quality and reliability of the proposed benchmark without evaluation from medical experts?
- How would medical professionals perform on the benchmark, and how the performance of AI models compares to human performance?
- How is it supported in the work that models *need* to integrate information across frames in order to answer the questions correctly?
- Please clarify the rationale behind the difficulty filtering step.

Minor:
- What does "# Rate" refer to in table 1?

---

### Official Review · Reviewer_hVqG · 2025-10-23

**Soundness:** 2
**Presentation:** 3
**Contribution:** 2
**Rating:** 4
**Confidence:** 4

**Summary:**

This paper introduces MedFrameQA designed to test multi-image medical VQA, focusing on the type of comparative reasoning clinicians use when analyzing multiple related scans. The authors develop an automated pipeline that extracts and links coherent frames from educational medical videos from YouTube, generating VQA pairs that require reasoning across 2 - 5 related images. The final dataset includes 2851 QA pairs covering several body systems and organs. They evaluate leading MLLMs like GPT-4o, Gemini 2.5 Flash, Claude, Qwen and show that all models perform poorly because current MLLMs struggle with integrating evidence across images and maintaining coherent reasoning chains. The benchmark highlights the gap between current AI abilities and the multi-image diagnostic reasoning required in clinical practice.

**Strengths:**

- The benchmark is diverse and focuses on multi image clinical reasoning, which is currently underexplored in medical AI evaluations.
- The proposed pipeline automates the extraction of captions and the generation of multiple choice questions with clinical reasoning. The authors also evaluate several open and closed source models, showing that they underperform in this complex setting.

**Weaknesses:**

- This benchmark is built by curating relevant frames from YouTube videos with medical explanations. The images provided to the MLLMs are often slides containing screenshots of medical images with borders and annotations. This is not very indicative of real-world scenario where high resolution medical images are ingested by the MLLMs. For this reason, optimizing models for this benchmark may not translate well to practical clinical use cases.
- The multiple choice questions, answers and reasoning chains are generated using GPT-4o without any clinical validation. Although they are grounded in captions extracted from the videos, GPT-4o could still hallucinate information. Expert clinical validation of the final benchmark is therefore necessary to confirm the legitimacy of the synthesized samples.
- The difficulty filtering process described in Section 3.5 may introduce bias into the benchmark. Models like GPT-4o and o1 are used to filter out questions that were answered correctly, which could explain their lower performance on the filtered benchmark. If the same filtering process were applied using Gemini-2.5-Flash, its performance would likely decrease on the resulting benchmark as well.

**Questions:**

Please address the above weaknesses.
- Is there any filtration done for quality control when curating the YouTube videos? Perhaps based on subscriber count or video views?
- Were there any steps taken to crop out the medical images from the video frames? If so, how was this process done?
- Gemini 2.5 Pro supports image inputs, this model is not evaluated in the benchmark.
- In Table 1, The definition of Real World Scenarios is not clearly stated. How is that MedXpertQA MM falls under Real World Scenarios, but benchmarks like MMMU (H&M) and MMMU-Pro (H&M) does not fall under this category, even though they support multiple images with paired reasoning.

---

### Official Review · Reviewer_WBEu · 2025-10-31

**Soundness:** 3
**Presentation:** 3
**Contribution:** 3
**Rating:** 8
**Confidence:** 4

**Summary:**

This paper introduces MEDFRAMEQA, a medical VQA benchmark that focuses on clinical reasoning across multiple images. The data come from 3,420 medical education videos. The authors extracted 111,942 key frames and kept 9,237 high-quality frames, grouped into question sets of 2–5 images. In total there are 2,851 multiple-choice questions covering 9 systems, 43 organs, and 114 modality × finding keywords. Each question includes a gold-standard rationale aligned with the video transcript. The authors evaluate 11 multimodal models; most get below 50% accuracy, showing that current models struggle with multi-image reasoning.

**Strengths:**

The task is clearly defined and close to clinical practice. It explicitly requires cross-frame evidence aggregation (multi-view, multi-timepoint, cross-modality) within a single clinical case, rather than treating multiple images as loose, unrelated inputs. The comparison with single-image/non-video datasets like VQA-RAD, SLAKE, PMC-VQA, and OmniMedVQA is clear. Results demonstrate that even SOTA models perform low overall and vary widely across subsets on this benchmark. The data pipeline is also scalable.

**Weaknesses:**

Many steps rely on models like GPT-4o. Even with human checks, this creates a risk of distribution shift. Adding multi-source expert review or cross-model verification could help. The paper mentions human evaluation (including filtering items “devoid of significant visual medical content”), but it isn’t clear whether there was systematic review by radiology/clinical experts or agreement metrics. The final version should report who the annotators were, their process, and inter-rater agreement. Currently the questions are mostly single-choice. Consider adding evidence localization, multi-select, or open-ended generation with rationale-based scoring to improve constraints and interpretability.

**Questions:**

See Weaknesses

---

### Note · Authors · 2025-11-29

**Comment:**

We would like to thank the reviewers for their thoughtful comments and feedback.

**Withdrawal Confirmation:**

I have read and agree with the venue's withdrawal policy on behalf of myself and my co-authors.